# Electromagnetic Shielding Effectiveness of Woven Fabric with Integrated Conductive Threads after Washing with Liquid and Powder Detergents

**DOI:** 10.3390/polym14122445

**Published:** 2022-06-16

**Authors:** Tanja Pušić, Bosiljka Šaravanja, Krešimir Malarić, Marta Luburić, Tea Kaurin

**Affiliations:** 1Department for Textile Chemistry and Ecology, Faculty of Textile Technology, University of Zagreb, 10000 Zagreb, Croatia; tanja.pusic@ttf.unizg.hr (T.P.); tea.kaurin@ttf.unizg.hr (T.K.); 2Department for Clothing Technology, Faculty of Textile Technology, University of Zagreb, 10000 Zagreb, Croatia; martajarakm@gmail.com; 3Department of Communication and Space Technologies, Faculty of Electrical Engineering and Computing, University of Zagreb, 10000 Zagreb, Croatia; kresimir.malaric@fer.hr

**Keywords:** fabric, stainless-steel threads, shielding effectiveness (SE), washing, detergent

## Abstract

The paper investigates the shielding effectiveness of a newly developed cotton and polyester fabric into which conductive stainless-steel threads were incorporated in the warp and weft directions at frequencies 0.9 GHz, 1.8 GHz, 2.1 GHz, and 2.4 GHz. As resistance to external influences and degradation is an additional critical factor for protective textiles, the newly developed protective fabric was exposed to cumulative wash cycles with liquid and powder detergents, which were targeted to preserve the shielding effectiveness (SE). In addition to the SE shielding effectiveness, the influence of 10 washing cycles on the change in thickness as a structural parameter was analyzed. Micro-images of fabric surfaces before and after the first, third, fifth, seventh, and tenth washing cycles were also observed. The obtained results showed that powder and liquid detergents were well formulated to preserve the electromagnetic shielding effectiveness (EMSE) at higher frequencies. However, their impact on the appearance of the surface was not fully consistent with the shielding effectiveness.

## 1. Introduction

Shielding against electromagnetic (EM) interference is an important requirement for an effective degree of protection in technology, especially in the field of aircraft protection, aviation, and portable electronics [1]. Electrically conductive materials are widely used in different applications because they can affect human health and the environment [2].

Numerous techniques are available for the production of electrically conductive structures, such as layered metals, metal salt films, and conductive polymers on textile surfaces [3,4]. It is known that textile materials containing metal threads with good electrical conductivity, which can provide protection in workplaces where radiation sources can be hazardous to human health, are used to protect against radiation. Depending on the radiation source and exposure of the human body, workwear may have different protective properties depending on the composition of the textile material, thickness, type and amount of metal threads, additional finishing, etc.

Depending on the type of conductive textiles and their purpose, the durability of a certain functional property is evaluated. The change in appearance is a property that can be considered as an additional indicator, and in some cases, this indicator is directly related to functional properties [5,6].

Functional electromagnetic shield effectiveness EMSE textiles get dirty during use, so wash durability is a requirement that needs to be analyzed, and their use value, i.e., functional lifetime, needs to be analyzed. The assessment of washability depends on the purpose of the functional textiles, the frequency of washing, and the prediction of the number of cycles during the textile’s lifetime [7].

The washing process is characterized by four interdependent parameters: time/duration, mechanics, temperature, and chemistry, which are referred to as the Sinner’s factors [8]. Therefore, in this process, chemical action (detergent, surfactants), heat stress (washing/drying temperature), solvent (water), and mechanical stress (friction, abrasion, flexion, hydro-dynamic pressure, folding of clothing) are the potential factors that can damage the conductive components in textiles and impair the degree of protection and aesthetic properties [9]. The resistance of conductive textiles to washing is often assessed through the change in resistance, by varying the parameters of the washing (detergent, temperature, number of cycles, mechanics) and drying processes (temperature, time, environment) [10]. A high durability during washing and drying was displayed by conductive yarns coated with silicone and nano-silicone products, where the washing process was carried out according to ISO 6330: 2012 for 10 cycles in water at 30 °C with slight rhythmic rotation. The favorable results were due to the high fluidity and adhesive properties of the applied silicones that provided the threads with a thin coating, which protected them from chemical and mechanical action in the process of washing in water [11,12].

Interlock and single jersey knitted fabrics from a blend of cotton and stainless steel were subjected to the washing process with detergent according to ISO 6330 at 40 °C. After the five washing cycles, the interlock fabrics possessed better SE protection values than single jersey fabrics in the low and medium frequency range [13].

The anionic surfactant SDS (sodium dodecyl sulfate) and water have a minor and equal effect on the change in the resistance of yarn coated with poly (3,4-ethylenedioxythiophene): poly (4-styrenesulfonate) and post-treated with ethylene glycol (EG). By treatment with EG, conductive yarns maintained electrical conductivity and good stability in repeated washing and drying cycles [14,15].

It was proved that is possible to produce a patterned, electrically conductive fabric by digital printing. The shielding ability of such fabric was reduced by increasing cycles of washing at 30 °C with commercial universal detergent (pH 7.8) and drying [16].

Research on the electrical properties of fabrics that had integrated conductive polyester yarns layered with silver and gold particles in the weft direction and were subsequently treated with polyvinyl alcohol, starch, and fabric softener has proven that the conductive layer gets damaged during washing and drying, thus also impairing the protective properties [17].

The newly developed functional and conductive yarn with incorporated silver nanoparticles is woven into the fabric, thus achieving water-repellent and self-cleaning properties. This fabric retains excellent electrical resistance, mechanical properties, and stability through cyclic washing in a solution of commercial liquid detergent [18].

The service life of functional conductive threads should be equal to the fabric in which they are integrated [19]. The existing research on the washability of conductive textiles is not focused only on this property, it is mainly based on the protocol prescribed by ISO 6330:2021-Textiles-Domestic washing and drying procedures for textile testing, applying a variety of agents, from standard and commercial detergents to water and surfactants. The development of the chemical industry makes it possible to improve the formulation of care and maintenance products for protective textile materials as well as clothing made from them. Thus, formulated special agents for maintaining certain protective properties can contribute to a longer lifetime for protective clothing.

This paper analyzes the EMSE of the newly developed fabric with incorporated conductive threads in both directions before and after cyclic washing with special liquid and powder detergents targeted to retain the EMSE of functional textiles and monitored at frequencies 0.9 GHz, 1.8 GHz, 2.1 GHz, and 2.4 GHz.

## 2. Materials and Methods

### 2.1. Material

The research is focused on a newly designed fabric from cotton and polyester with stainless-steel threads in the warp and weft direction, produced on the weaving machine PICANOL OMNI PLUS 800 in the factory Čateks d.d. in Čakovec, Figure 1.

Fiber composition of basic warp and weft yarn is polyester 50% and cotton 50%. Stainless-steel threads woven in the warp and weft direction, after 38 warp and 20 weft threads, provided the EMSE of fabrics. The composition of effective warp and weft is cotton 90% and stainless-steel thread (trade name Bekinox) 10%.

The characteristics of the fabric designed in this way are shown in Table 1.

The effect of electromagnetic shielding, EMSE, was checked before and after cyclic washing with special detergents in liquid and powder formulations.

The fabric was exposed to a cyclic washing process with special liquid and powder detergents, both supplied by YShield, the formulations of which are designed to retain EMSE. The composition of the detergents is specified in accordance with EC Regulation 648/2004.

The liquid detergent Texcare (LD) contains potassium soap (5–15%), nonionic surfactant alkyl polyglucoside C8-16 (5–15%), anionic surfactant sodium C8-14 fatty alcohol sulfate (5–15%), ethanol (1–5%), potassium citrate (<1%), sulfated castor oil, and water to fill the volume to 100%.

The powder detergent Texcare (PD) contains zeolite (15–30%), coco glucoside (5–15%), citrate (15–30%), sodium soap (15–30%), sodium carbonate (5–15%), silicates (1–5%), sodium bicarbonate (1–5%), Quillaja saponaria as the source of saponins (<1%), and water to fill the volume to 100%.

### 2.2. Methods

The protective fabric was exposed to cyclic washing processes (W) with liquid and powder detergents at a concentration of 3.85 g/L in the laboratory device Polymat Mathis with a bath ratio of 1:10 at 30 °C for 30 min through 10 cycles. After each washing cycle, three rinsing cycles with distilled water (DW) were performed. After rinsing, the fabrics were centrifuged using Koh and Nor spin dryer for 3 min and air-dried.

In the research, analytical methods were applied for the characterization of detergent solutions through the measurement of pH and conductivity.

Additionally, non-destructive methods were selected for fabric characterization before and after the washing cycle, the microscopic method for surface inspection, and the thickness measurement.

Fabric surface characterization before and after the first, third, fifth, seventh, and tenth washing cycles with powder and liquid detergent was performed with the optical digital microscope, DinoLite LT Premier IDCP B.V., Almere, the Netherlands, under 235× magnification.

The thickness of the fabric before and after 10 washing cycles was determined in accordance with EN ISO 5084:2003-Textiles-Determination of thickness of textiles and textile products.

Changes in the thickness (dh) of the fabric after 10 washing and drying cycles compared to the initial thickness were obtained according to expression 1 [20].
dh = (h_1_ − h_2_)/h_2_ × 100 (%)(1)
h_1_—thickness of pristine fabric (mm); h_2_—thickness of washed fabric (mm).

The SE protective properties of fabrics before and after the first, third, fifth, seventh, and tenth washing cycles were investigated by a method developed at the Department of Communication and Space Technologies, University of Zagreb, Faculty of Electrical Engineering and Computing, at a temperature of 23 ± 1 °C and relative humidity of 50 ± 10%.

The measuring setup was designed according to the recommendations of international standards IEE-STD 299-97, MIL STD 285, and ASTM D-4935-89, and consists of the measuring instrument NARDA SRM 3000, signal generator HP 8350 B, IEV antenna horn—telecommunications industry, Ljubljana, Type A12 and a frame of 1 m × 1 m (in which the samples are placed), Figure 2 and Figure 3.

The NARDA SRM 3000 measuring instrument (Figure 3a) is a portable spectrum analyzer that uses a probe to measure the field isotropically, i.e., from any direction and any polarization, which makes the measurement easier. It is intended for measuring EM radiation in the frequency range from 80 to 3000 MHz, where it gives an extremely linear response. The measurement should be performed within the above operating conditions, and the instrument must be calibrated regularly to show accurate values of the measuring field.

The HP 8350 B signal generator, Figure 3b, is an instrument that serves as a source of electromagnetic radiation. A continuous sine wave generator at the frequencies 0.9 GHz, 1.8 GHz, 2.1 GHz, and 2.4 GHz was used for testing [14]. To obtain higher values, the sine wave generator must sometimes be used in conjunction with a microwave amplifier to obtain higher electromagnetic field values.

The horn antenna (Figure 3b) was named after its shape and is used to receive and transmit microwave signals. When receiving, it is used to collect and direct radio waves towards the waveguide, and when transmitting, it is used to direct radio waves from the waveguide into space. For greater antenna gain, the horn should have a larger opening. The inner sides of the antenna are made of a conductive material such as copper, brass, silver, aluminum, etc. [14]

A frame with a wooden stand was made for the purpose of testing the electromagnetic shielding effectiveness (Figure 3c). A frame of 1 m × 1 m consists of the front and back side of the frame between which the test sample is placed, and after which the frames are joined by holders to keep the textile sample fixed during measurement. During the measurement, the frame with the sample was inserted into a wooden stand to keep it in a vertical position. The shielding effectiveness (SE) is calculated according Equation (2):SE = 20 logE_0_/E_1_(2)
where E_0_ is the level of received electric field without a shield and E_1_ is the received level of electric field with a shield.

## 3. Results

Liquid and powder detergents were formulated in such a way as to retain the protective properties of fabrics with conductive elements, e.g., coated with silver and with integrated conductive yarn, e.g., stainless steel. As previously specified, the composition of the two detergents differs, primarily in the type and mass ratio of the surface-active substances (surfactants). The surface-active substances in the liquid detergent, represented by a mass ratio of 15–45%, were soap, non-ionic surfactant, and anionic surfactant.

According to its composition, it can be estimated that this liquid detergent is structured with an equal ratio of surfactants: anionic, non-ionic, and soap. Such detergents contain solvents, alcohols, and a large amount of soap with a certain length of hydrocarbon chain to soften the water so that the total content of active components can be up to 60%; it also contains citrate as an important component in the liquid detergent formulation [21,22].

The surface-active substances in the powder detergent, present in a mass ratio of 20–45%, were soap and the non-ionic surfactant alkyl polyglycoside (APG). The composition was enriched with a small mass ratio of Quillaja saponaria (<1%) as a source of saponins, i.e., natural surfactants [23,24,25].

Based on the presented composition, both detergents have an environmentally friendly profile and benefit due to the APG into the formulations. This surfactant belongs to the group of environmentally friendly non-ionic surfactants due to its high degree of biodegradability, low aquatic toxicity, origin from renewable sources, favorable dermatological properties, and mildness to the skin and eyes [26].

It is important to consider the properties of detergent solutions by measurement of the pH and conductivity, Table 2.

The results in Table 2 indicate the differences in pH and conductivity of the liquid and powder detergent solutions. The obtained values are completely in accordance with the composition, according to which the liquid detergent has a pH of 8.05 and the powder detergent a pH of 10.30. Sodium salts (carbonate, bicarbonate, silicates and zeolite) are carriers of alkalinity and are responsible for the high alkalinity of the powder detergent.

The differences in the conductivity of the detergent solutions in relation to the conductivity of water are also different, with the components of liquid detergent increasing the conductivity by 250 µS/cm and of the powder detergent by 1.680 µS/cm. These differences suggest that the powder detergent solution is a more complex dispersion system than the liquid one.

The designed fabric with integrated electrically conductive stainless-steel threads consists of a cotton/polyester blend. Given the ratio of cotton, which has high swelling in aqueous solutions, shrinkage in detergent solutions is expected during washing. Table 3 shows the differences in the thickness of the washed fabrics in relation to the reference pristine fabric, with thickness of 0.58 mm.

The results of the change in thickness of the washed and dried samples calculated according to equation 1 show an increase in the thickness of the fabric in the cumulative washing processes with liquid and powder detergents. The thickness of the fabric washed with liquid detergent after 10 cycles was slightly greater than the one washed with powder detergent. The obtained results regarding the changes in thickness are similar to the results of other researchers despite the differences in the structural parameters of the studied textiles [20,27].

The appearance of the fabric surface is a property that can indicate the degree of damage to the electrically conductive threads so that the evaluation of the effects of cyclic washing with liquid and powder detergents on the degree of protection includes surface analysis with a digital optical microscope. Table 4 shows the images of the untreated sample and the samples after the first, third, fifth, seventh, and tenth washing cycles with liquid and powder detergents. Micro-images of the fabric surface were taken at a 235× magnification.

In Table 4, the image of the untreated sample indicates compactness and uniform alternation of warp and weft threads. The images of the fabric surface after washing with liquid detergent show good durability of the fabric structure after an increased number of washing cycles. Fibrils are noticeable only after the seventh cycle, which can be attributed to the cotton component, which has a fibrillation tendency in the washing process. The warp and weft threads of the fabric are slightly intricate, protruding after 10 washing cycles, and a slight disorientation of the surface is visible.

Micro-images of the fabric washed with powder detergent indicate differences in the compactness of the structure, where the warp and weft threads are slightly deformed, and protruding fibers are more noticeable, i.e., their number is higher compared to the sample washed with liquid detergent.

The EMSE of the designed fabrics with electrically conductive threads incorporated into the structure in the warp and weft direction before and after the first, third, fifth, seventh, and tenth washing cycles with powder and liquid detergents was determined at frequencies of 0.9 GHz, 1.8 GHz, 2.1. GHz, and 2.4 GHz (Table 5 and Table 6; Figure 4 and Figure 5).

Figure 4 shows the SE of the stainless-steel fabric sample (face) before and after the liquid detergent washing cycle.

According to the results, there is a linear drop in SE at all frequencies for the face of the tested fabric washed with liquid detergent. At 0.9 GHz, the total drop in SE after the 10th washing cycle was 3.28 dB. At 1.8 GHz the difference was 3.8 dB, and at 2.1 GHz the drop was 4.64 dB. The SE drop after 10 processing cycles at 2.4 GHz was 6.17 dB.

Figure 5 shows the SE of the stainless-steel fabric sample (face) before and after the powder detergent washing cycles.

Table 6 and Figure 5 show the decrease in SE, which, according to the corresponding graph, shows a deviation in the results of SE for the face of the tested fabric washed with powder detergent. At a frequency of 0.9 GHz, the difference between the SE values of the first and the tenth washing cycle was 5.59 dB. At 1.8 GHz, the SE drop was slightly more linear and the difference was 4.11 dB. A slightly larger deviation in the SE results was noticeable at the frequency of 2.1 GHz, where the difference between the SE of the initial untreated sample and the same sample after the tenth washing cycle was 5.04 dB. At 2.4 GHz, the SE drop after the first washing cycle was significantly higher, with a difference of 3.45 dB. The total drop in SE after 10 washing cycles, relative to the untreated sample, was 6.73 dB.

Table 7 shows the SE values of fabric samples before and after 10 washing cycles with liquid and powder detergents measured at frequencies 0.9 GHz, 1.8 GHz, 2.1 GHz, and 2.4 GHz.

The results in Table 7 show that the EMSE of fabrics washed with liquid and powder detergents over 10 washing cycles, relative to the unwashed sample, depend on the frequency. Liquid detergent has less of an effect on changing the degree of protection at the frequency of 0.9 GHz compared to powder detergent. These differences in the degree of protection from EM fields at 1.8 GHz and 2.1 GHz are insignificant. At 2.4 GHz, the SE properties of fabrics washed with liquid detergent were somewhat better preserved than those washed with powder detergent. It is interesting to monitor the influence of 10 washing cycles with liquid and powder detergent on the change in the EMSE of fabric at the frequency of 1.8 GHz, 2.1 GHz, and 2.4. GHz. The SE degree of the materials with incorporated stainless-steel threads in the warp and weft direction increases with the increase in frequency at 1.8 GHz (14.33 dB), 2.1 GHz (15.58 dB), and 2.4. GHz (17.60 GHz). Despite these differences in the SE of the initial fabric, after 10 washing cycles with liquid and powder detergents, it was almost uniform and reached approximately 10 to 11 dB. According to [16], the obtained results indicate good grading of electromagnetic shielding fabric for general use.

After 10 cycles, the fabric washed with both detergents maintained the same level of protection at the frequencies of 2.1 GHz to 2.8 GHz due to its structural properties. The newly developed fabric, within the scope of this research, possesses EMSE values at high frequencies, so it can be used as electromagnetic shielding material. Future studies are needed to improve the EM shielding properties of fabrics with integrated stainless-steel threads at low frequencies as they correspond to the study [28,29].

## 4. Conclusions

The newly designed cotton–polyester fabric with incorporated stainless-steel conductive threads, woven in regular segments in the direction of the warp and weft threads, has EMSE properties at the frequencies of 0.9 GHz, 1.8 GHz, 2.1 GHz, and 2.4 GHz. The SE increases with the increase in frequency. The protocol of the washing process is important if the added value of the protective textiles is to be maintained. The protective fabric was analyzed at the same frequencies after cyclic washing at 30 °C with liquid and powder detergents specially formulated to retain the SE properties. The results proved that, with after cyclic washing, the degree of protection decreases, and its value depends on the frequency. The greatest influence of washing on the decrease of the SE properties was recorded at the frequency of 0.9 GHz, where powder detergent had a stronger influence than liquid detergent. The preservation of the protective properties of fabrics cyclically washed with both detergents is almost the same despite the difference in the initial SE values at the frequencies of 1.8 GHz, 2.1 GHz, and 2.4 GHz. Despite the differences in the composition and properties of the analyzed detergents, based on these results, it can be concluded that the powder and liquid detergents are well formulated if the EMSE is monitored at higher frequencies. However, the impact of detergents on the appearance of the fabric surface is not fully consistent with its protective properties. Namely, the micro-images of the washed samples revealed a greater impact of the powder detergent on the fabric surface than did the liquid one. Fabric samples washed with liquid detergent had good surface and structural stability.

## Figures and Tables

**Figure 1 polymers-14-02445-f001:**
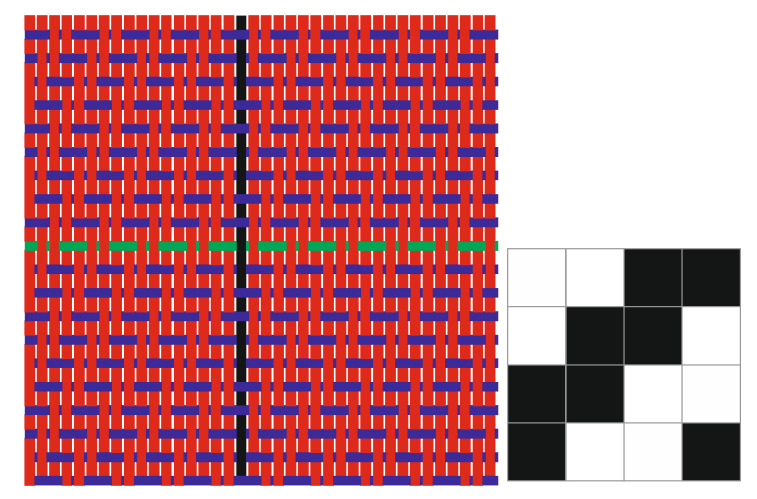
Graphic presentation of thread interlacing for twill fabric. Red = basic warp threads (cotton/PES); Blue = basic weft threads (cotton/PES); Black = effective warp thread (Bekinox); Green = effective weft thread (Bekinox).

**Figure 2 polymers-14-02445-f002:**
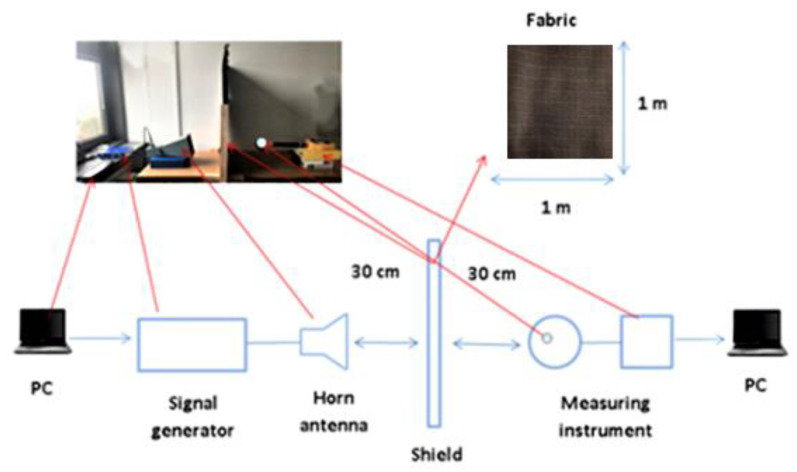
Schematic view of measuring set-up created by authors.

**Figure 3 polymers-14-02445-f003:**
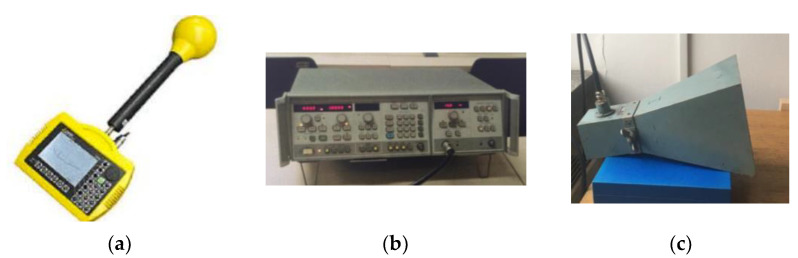
Measuring set-up: (**a**) measuring instrument NARDA SRM 3000; (**b**) HP 8350 B signal generator; (**c**) horn antenna.

**Figure 4 polymers-14-02445-f004:**
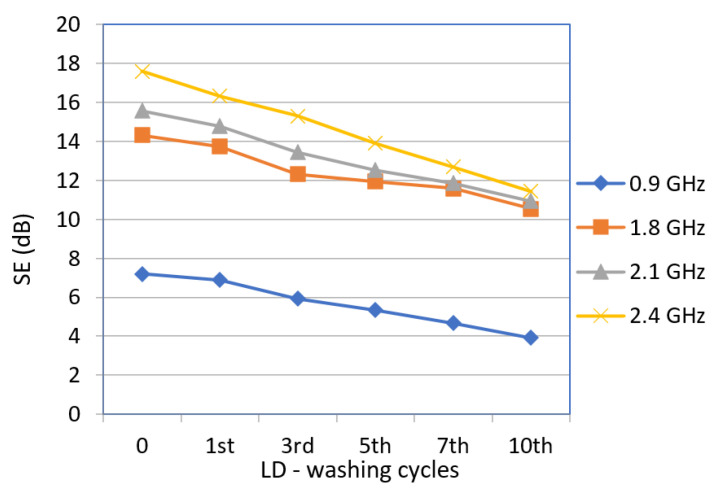
Shielding effectiveness (SE) of the fabric sample with stainless steel (face) before and after the 1st, 3rd, 5th, 7th, and 10th washing cycles with liquid detergent at frequencies 0.9 GHz, 1.8 GHz, 2.1 GHz, and 2.4 GHz.

**Figure 5 polymers-14-02445-f005:**
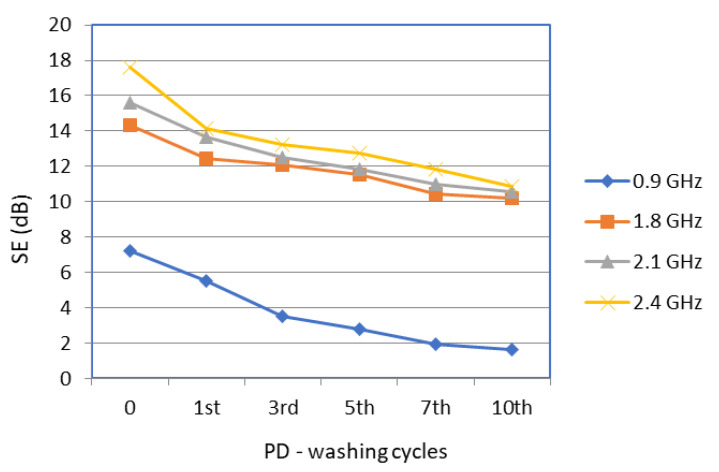
Shielding effectiveness (SE) of stainless-steel fabric sample (face) before and after the 1st, 3rd, 5th, 7th, and 10th washing cycles with powder detergent at frequencies 0.9 GHz, 1.8 GHz, 2.1 GHz, and 2.4 GHz.

**Table 1 polymers-14-02445-t001:** Structural properties of the stainless-steel fabric.

Micrograph, Magnification 50×	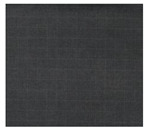
Weave	Twill 2/2
Composition	49% PES, 48% cotton, 3% inox
Surface area (g/m^2^)	249
Fineness of warp, weft, and inox threads (tex)	20
Density in warp direction (threads/10 cm)	380 + 10 (inox)
Density in weft direction (threads/10 cm)	200 + 10 (inox)

**Table 2 polymers-14-02445-t002:** pH and conductivity of detergent solutions.

Detergent	pH	T (°C)	κ (µS/cm)	T (°C)
LD	8.05	17.2	267.2	17.9
PD	10.30	17.0	1715.0	
DW	6.20	17.2	4.9	17.2

**Table 3 polymers-14-02445-t003:** Change in fabric thickness after 10 washing cycles in liquid and powder detergents.

Sample	dh (%)
Pristine	
W_LD 10	−12.12
W_PD 10	−10.76

**Table 4 polymers-14-02445-t004:** Characterization of fabric surface before and after washing with liquid and powder detergents.

Sample	Micrograph, 235×	
Pristine	** 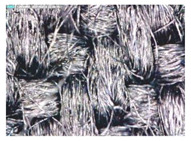 **	
Washing cycles	LD	PD
1st	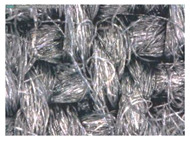	** 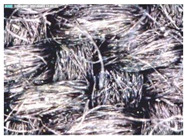 **
3rd	** 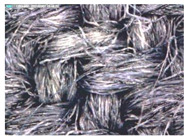 **	** 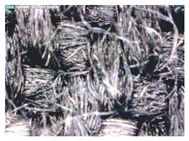 **
5th	** 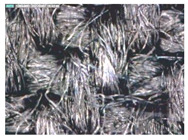 **	** 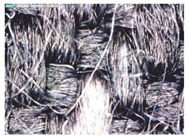 **
7th	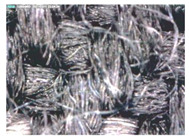	** 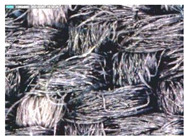 **
10th	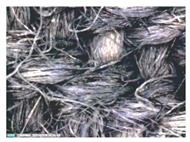	** 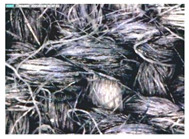 **

**Table 5 polymers-14-02445-t005:** Shielding effectiveness (EMSE) of stainless-steel fabric sample (face) before and after the 1st, 3rd, 5th, 7th, and 10th cycles of washing with liquid detergent at frequencies 0.9 GHz, 1.8 GHz, 2.1 GHz, and 2.4 GHz.

Samples	0.9 GHz	1.8 GHz	2.1 GHz	2.4 GHz
	SE (dB)			
Pristine	7.20	14.33	15.58	17.60
W_LD 1th	6.88	13.75	14.77	16.34
W_LD 3rd	5.93	12.33	13.45	15.30
W_LD 5th	5.35	11.93	12.52	13.90
W_LD 7th	4.68	11.58	11.86	12.69
W_LD 10th	3.92	10.53	10.94	11.43

**Table 6 polymers-14-02445-t006:** Shielding effectiveness (EMSE) of sample of stainless-steel fabric (face) before and after the 1st, 3rd, 5th, 7th, and 10th washing cycles with powder detergent at frequencies 0.9 GHz, 1.8 GHz, 2.1 GHz, and 2.4 GHz.

Samples	0.9 GHz	1.8 GHz	2.1 GHz	2.4 GHz
	SE (dB)			
Pristine	7.20	14.33	15.58	17.60
W_PD 1st	5.52	12.47	13.64	14.15
W_PD 3rd	3.52	12.06	12.48	13.23
W_PD 5th	2.77	11.54	11.83	12.74
W_PD 7th	1.91	10.47	10.98	11.85
W_PD 10th	1.61	10.22	10.54	10.87

**Table 7 polymers-14-02445-t007:** Shielding effectiveness (SE) of stainless-steel fabric sample (face) before and after 10 cycles of washing with liquid and powder detergent at frequencies 0.9 GHz, 1.8 GHz, 2.1 GHz, and 2.4 GHz.

Sample	0.9 GHz	1.8 GHz	2.1 GHz	2.4 GHz
	SE (dB)			
Pristine	7.20	14.33	15.58	17.60
W_LD 10	3.92	10.53	10.94	11.43
W_PD 10	1.61	10.22	10.54	10.87

## Data Availability

The data presented in this study are available on request from the corresponding author.

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
