# Peer review of "Electromagnetic Shielding Effectiveness of Woven Fabric with Integrated Conductive Threads after Washing with Liquid and Powder Detergents"

_polymers, 2022, doi:10.3390/polym14122445_

Round 1
Reviewer 1 Report
The paper entitled “Electromagnetic Shielding Effectiveness of Woven Fabric with Integrated Conductive Threads After Washing with Liquid and Powder Detergent” investigates the shielding effectiveness of a newly developed cotton and polyester fabric, the textiles exhibit good electromagnetic shielding properties as well as structural stability after washing cycles. The work is interesting and meaningful; thus, the manuscript is recommended for publication in “Polymers” after major revisions:
1. There are several grammatical mistakes in this article, which although do not affect readers' understanding, please revise them.
2. For equation 2, it should be corrected by using SE=20 logE0/E1.
3. On page 7, lines 221-225, the author should give the conductivity of water. In addition, the change in conductivity after adding LD and PD should be recalculated.
4. The authors give the SEM and ES analysis of the sample, the following articles maybe helpful to help the authors to further analyze this. https://doi.org/10.1016/j.cej.2022.136033; https://doi.org/10.1002/EXP.20210029; https://doi.org/10.1016/j.cej.2021.131608; https://doi.org/10.1002/EXP.20210152; https://doi.org/10.1002/EXP.20210131; https://doi.org/10.1002/EXP.20210178.
Author Response
Dear Reviewer 1
Authors are grateful for suggestions and comments for improvement of a manuscript!
Authors

Reviewer 2 Report
The submitted manuscript entitled "Electromagnetic shielding effectiveness of woven fabric with integrated conductive threads after washing with liquid and powder detergent "is focused on the analysis of electromagnetic shielding effectivity of developed cotton/polyester fabric with the conductive steel threads.
The comments and requests:
- - Could you please specify the effects of electrically conductive materials on human health and the environment in the introduction?
- - Could you please add to the introduction the values of shielding efficiency for different types of materials from the literature? In the discussion, you state the measured values of the researched materials. Still, the reader should be able to compare the effectiveness of your developed material with the values of other materials.
- - In table 1 is a micrograph that is entirely invisible. What structure should be there?
- - Please describe both detergent's content in the Materials part. Procedure (2.2.), please give to the Methods part.
- - In Table 3, you listed the fabric thickness after 10 washing cycles. What was the thickness of the pristine sample?
- - I do not understand the statement: "The results show an increase in the thickness of the fabric in the cumulative washing processes with liquid and powder detergent, where the thickness of the fabric washed with liquid detergent after 10 cycles was slightly greater than the one washed with powder detergent ". Because in the mentioned Table 3, there are the thicknesses negative. Could you explain it?
- - Could you please discuss your results with the literature? Are the values of SE for investigated fabric high, low, or expected? What is the SE of the fabrics/samples which other authors already investigated?
- - What is the electromagnetic radiation in the applications you mention, such as aviation or portable electronics?
I recommend the major revision of the manuscript.
Author Response
Dear Reviewer 2
Authors are grateful for suggestions and comments for improvement of a manuscript!
Authors

Reviewer 3 Report
Comments to the author
Title: Electromagnetic Shielding Effectiveness of Woven Fabric with Integrated Conductive Threads After Washing with Liquid and Powder Detergent
ID_polymers-1747329
The article lies on the scope of the journal and well structured, however, the following issues must be addressed before considering publication
1. Since there are lots of papers in this area, it is difficult to consider as a new product: refer: https://www.mdpi.com/1996-1944/15/8/2892 and https://core.ac.uk/download/pdf/229215642.pdf
2. The conductivity measurement must be conducted
3. The electromechanical (Stress-strain-conductivity) must be addressed: Refer: https://link.springer.com/article/10.1007/s10853-019-03519-3
4. How come a good result then if the consistency against the surface is not good. any recommendation
5. Introduction is too much
6. EMSE measurement must be conducted co-axially
7. What is the effectiveness of EMSE in terms of the content of conductivity?
8. Show the effect of humidity on EMSE
9. What is the effect of increasing the content of stainless steel yarn
Author Response
Dear Reviewer 3
Thanks for comments and suggestions for improvement of a manuscript!
Authors

Round 2
Reviewer 2 Report
The questions were answered and the ambiguities clarified. I recommend to accept the manuscript.
Reviewer 3 Report
well answered